# Peptide-Mediated Gene Transfer into Marine Purple Photosynthetic Bacteria

**DOI:** 10.3390/ijms21228625

**Published:** 2020-11-16

**Authors:** Mieko Higuchi-Takeuchi, Takaaki Miyamoto, Choon Pin Foong, Mami Goto, Kumiko Morisaki, Keiji Numata

**Affiliations:** 1Biomacromolecules Research Team, RIKEN Center for Sustainable Resource Science, 2-1, Hirosawa, Wako-shi, Saitama 351-0198, Japan; takaaki.miyamoto@riken.jp (T.M.); mami.goto.wj@riken.jp (M.G.); rabotwo@ezweb.ne.jp (K.M.); 2Department of Material Chemistry, Graduate School of Engineering, Kyoto University, Kyoto-Daigaku-Katsura, Nishikyo-ku, Kyoto 615-8510, Japan; foong.choonpin.8a@kyoto-u.ac.jp

**Keywords:** transformation, plasmid DNA delivery, *Rhodovulum sulfidophilum*, cell penetrating peptide

## Abstract

Use of photosynthetic organisms is one of the sustainable ways to produce high-value products. Marine purple photosynthetic bacteria are one of the research focuses as microbial production hosts. Genetic transformation is indispensable as a biotechnology technique. However, only conjugation has been determined to be an applicable method for the transformation of marine purple photosynthetic bacteria so far. In this study, for the first time, a dual peptide-based transformation method combining cell penetrating peptide (CPP), cationic peptide and Tat-derived peptide (dTat-Sar-EED) (containing D-amino acids of Tat and endosomal escape domain (EED) connected by sarcosine linkers) successfully delivered plasmid DNA into *Rhodovulum sulfidophilum,* a marine purple photosynthetic bacterium. The plasmid delivery efficiency was greatly improved by dTat-Sar-EED. The concentrations of dTat-Sar-EED, cell growth stage and recovery duration affected the efficiency of plasmid DNA delivery. The delivery was inhibited at 4 °C and by A22, which is an inhibitor of the actin homolog MreB. This suggests that the plasmid DNA delivery occurred via MreB-mediated energy dependent process. Additionally, this peptide-mediated delivery method was also applicable for *E. coli* cells. Thus, a wide range of bacteria could be genetically transformed by using this novel peptide-based transformation method.

## 1. Introduction

Genetic transformation is the process of introducing exogeneous DNA into cells and is an important method in bioengineering technology. Natural transformation, conjugation and transduction are general methods for transformation in bacteria. In the process of conjugation, DNA is transferred from donor cell to recipient cell. Transduction is the process that bacteriophage mediates the transfer of DNA. Natural competence is the ability of bacteria to uptake of DNA from the environment and has been used transformation among many bacteria species [1,2]. The competency is known to be induced by chemical treatment such as divalent cations mainly calcium ion [3,4]. Method of chemically competent *E. coli* cells using calcium chloride has been developed by Hanahan [5] and is shown to be applicable for a lot of bacterial species [6,7]. Although the precise mechanism of the transformation induced by calcium chloride has not been solved yet, it is thought that divalent cations interact with negatively charged DNA and cell membranes, leading to DNA internalization into the cells due to alteration in membrane permeability [8]. Another method for induction of competence in bacteria is the electroporation. Electroporation is a physical method that uses electrical pulse to create temporary pores in cell membranes, allowing DNA internalization [9]. Competent cells need to be prepared in both calcium chloride and electroporation methods, and the preparation steps of competent cells are time consuming.

Recently, various types of nanomaterial-based gene delivery have been developed. One of the gene carrier systems is protein- and peptide-based materials [10]. Self-assembled peptide and protein nanomaterials such as viral capsids and nanofibers have great potential in various aspects of biomedical engineering [11]. Nucleic acids have the ability to self-assemble into compact structures. Self-assembled structure composed of cationic peptides and negatively charged DNA can be useful for efficient gene delivery applications. Carrier peptides have been designed composed of cell penetrating peptides (CPPs) and polycation amino acids (lysine and histidine) and used for bimolecular delivery into plant cells. CPPs have the ability to translocate across the cell membrane and polycation amino acids interact with the negatively charged DNA via ionic interactions to form complexes. These peptide-based carriers could deliver various types of molecules, such as DNA [12], RNA [13] and even protein, into plant cells [14,15]. Recently, the novel carrier peptides dTat-Sar-EED4 and dTat-Sar-EED5 were reported [16]. The dTat-Sar-EED peptides are composed of a retro-inverso Tat (dTat), which is one of the CPPs, a hexamer of sarcosine (Sar) and an endosomal escape domain (EED) [17]. The dTat-Sar-EED peptides were able to deliver biomolecules efficiently in a short period into plant and algal cells. Interestingly, a combination of peptide-plasmid DNA (pDNA) complex and dTat-Sar-EED peptide enhanced the delivery efficiency into plant cells [18]. On the other hand, the effect of dTat-Sar-EED peptides on the pDNA delivery into bacteria species has not been investigated so far.

Purple photosynthetic bacteria have only one photosystem and are considered to be the most primitive photosynthetic organisms. Purple photosynthetic bacteria are one of the ideal hosts as sustainable production [19,20], because they are known to have the ability for nitrogen fixing [21] as well as CO_2_ fixation, meaning that they can use N_2_ and CO_2_ in the air as nitrogen and carbon sources for their growth. Marine microorganisms have attracted attention as sources of genetic materials and bioactive metabolites [22]. Moreover, the use of marine microorganisms and seawater has great potential for the production of value-added chemicals from the viewpoint of lowering costs [23,24]. Several species of marine purple photosynthetic bacteria that can produce polyhydroxyalkanoate (PHA) biopolyesters have been reported [25,26,27]. In addition, hydrogen production [28], extracellular nucleic acid production [29], and spider silk production [30,31] have been studied using marine purple photosynthetic bacteria as host bacteria. Conjugation has been widely used for transformation in marine purple photosynthetic bacteria because electroporation method is not suitable for marine microorganisms due to high salinity. However, the conjugation method requires multiple steps and special plasmid. Recently, the preparation of chemically competent cells for marine purple photosynthetic bacteria by calcium chloride treatment was succeeded [32]. Although natural transformation is another means of bacterial transformation, to the best of our knowledge, it has not been established in marine purple photosynthetic bacteria.

In this study, the dual peptide-based delivery system to marine purple photosynthetic bacteria was examined. The pDNA complexes prepared by BP100-(KH)_9_ peptide and plasmid DNA carrying kanamycin resistance gene were introduced into the dTat-Sar-EED treated cells, resulting in the recovery of kanamycin resistant colonies. The pDNA delivery was largely inhibited at 4 °C and in the presence of A22 which is an inhibitor of the actin homolog MreB. Thus, pDNA has been successfully transduced into marine purple photosynthetic bacteria in a MreB-mediated energy dependent manner. 

## 2. Results and Discussion

### 2.1. Characterization of Peptide-Plasmid DNA Complex

One of the representative marine purple photosynthetic bacteria, *Rhodovulum sulfidophilum* (*R. sulfidophilum*), was used for pDNA delivery. The procedure for plasmid DNA delivery into *R. sulfidophilum* cells is shown in Figure 1. The BP100-(KH)_9_/plasmid DNA complex exhibited higher transfection efficiency compared to the (KH)_9_/pDNA complex in plant cells [18], indicating that CPP fusion enhance the pDNA delivery. Therefore, BP100-(KH)_9_ was used for the delivery into marine purple photosynthetic bacteria. The plasmid pBBR1MCS-2 [33] carrying kanamycin resistance gene (*KanR)* was used for pDNA delivery into *R. sulfidophilum* cells. 

The pDNA complexes were prepared at various N/P ratios ranging from 0.1 to 10 (the ratio of the moles of cationic amine groups of the peptides to those of phosphate groups of the DNA), with a fixed amount of pDNA (1 μg) and analyzed by electrophoretic mobility assay (Figure 2a). The pDNA migration was slightly retarded at higher N/P ratio and the DNA band disappeared at N/P ratio 10.0, indicating the stable formation of pDNA complexes. The average hydrodynamic diameter size and polydispersity index (PDI) of pDNA complex were analyzed by dynamic light scattering (DLS) (Figure 2b,c). The pDNA complexes at N/P 0.1 exhibited bimodal distributions and high PDI, suggesting heterogeneous pDNA complex formation. The diameter and PDI of the pDNA complexes at more than 0.5 were around 200 nm and 0.35, respectively. The zeta potential was high at N/P ratio of 5.0 (38.9 ± 1.0 mV) compared to N/P ratio of 0.5 (15.4 ± 0.2 mV). These properties of pDNA complexes were comparable to previous study [18]. 

### 2.2. Effects of dTat-Sar-EED5 Peptide on pDNA Delivery and Cell Viability

The *R. sulfidophilum* cell cultures (1.5 mL) were washed and resuspended in water before added with dTat-Sar-EED5 (Figure 1). Substitution with water from growth medium was important, because the growth medium contained high concentrations of NaCl (2%), leading to disturbance of ionic pDNA complex formation. The pDNA complex showed bimodal distribution in growth medium and average size was quite large (916 ± 40 nm). In addition, the pDNA complex in growth medium exhibited negative zeta potential (−18.9 ± 1.6 mV), which reported low gene delivery efficiency in plant cells [12].

Kanamycin resistant colonies were found after several days of cultivation (approximately 5 days). The presence of pDNA from kanamycin resistant colonies was verified by pDNA extraction followed by restriction digestion with Bgl*II* and Hind*III* (Appendix A), indicating that pDNA was delivered into *R. sulfidophilum* cells using dTat-Sar-EED5. Colony forming unit (CFU) was low, at 0 and 10 μM of dTat-Sar-EED5, and increased at 50 and 100 μM (Figure 3a), indicating that dTat-Sar-EED5 enhanced the delivery efficiency. On the other hand, the efficiency decreased at 300 μM of dTat-Sar-EED5 and resistant colony was not found at 500 μM (Figure 3a), suggesting the cytotoxicity effect of dTat-Sar-EED5 on cells. Actually, *R. sulfidophilum* culture became transparent just after the treatment at 500 μM of dTat-Sar-EED5 (Figure 3b), suggesting that cells were lysed by high concentrations of dTat-Sar-EED5. Evans blue staining assay was carried out to check the viability of *R. sulfidophilum* cells. The toxicity of dTat-Sar-EED5 was found in a concentration-dependent manner (Figure 3c). Cells were highly stained with Evans blue at 500 μM dTat-Sar-EED5, indicating that high concentrations of dTat-Sar-EED5 was toxic on R*. sulfidophilum* cells. High concentrations of CPPs affected cell viability because of changes in membrane integrity and inhibition of biosynthesis pathway [34,35]. Cytotoxic effect was also observed in plant cells at high concentrations of dTat-Sar-EED peptides [16]. On the other hand, both the delivery efficiency (Figure 3a) and the intensity of Evans blue staining (Figure 3c) were low at 10 μM dTat-Sar-EED5. The 10 μM dTat-Sar-EED5 might be below the threshold for induction of the delivery process. Accordingly, the optimum range of dTat-Sar-EED5 for pDNA delivery into *R. sulfidophilum* cells was 50–100 μM. 

### 2.3. Optimization of Plasmid DNA Delivery

The N/P ratio of the pDNA complex is reported to affect the delivery efficiency in plant cells [12]. To optimize the delivery condition, the pDNA delivery was examined at different N/P ratio (Figure 4a). The concentration of dTat-Sar-EED5 was fixed to 100 μM. Transformant was not recovered at N/P ratio of 0.1. The efficiency was high at higher N/P ratio compared to low N/P ratio. As shown in Figure 2, stable pDNA complexes were formed at higher N/P ratio. These results indicate that stable formation of pDNA complex result in efficient pDNA delivery into *R. sulfidophilum* cells. 

Next, effects of the recovery duration and cell growth stage on the delivery efficiency were examined (Figure 4b). Usually, the recovery step is needed to allow expression of antibiotics resistance gene in the process of transformation. When the recovery period without kanamycin was 3 h, the efficiency was greatly decreased compared to overnight incubation. Cells in the log phase are widely used for bacterial transformation. As shown in Figure 4b, the efficiency was largely decreased using steady state cells compared to log phase cells. Thus, these results indicate that long recovery period and actively proliferating cells are required for efficient pDNA delivery into *R. sulfidophilum* cells. Overall, the highest plasmid delivery efficiency was 600 CFU per µg of plasmid, which was obtained under condition with N/P ratio 2.0, 100 μM dTat-Sar-EED5, log phase cell culture and overnight recovery period.

### 2.4. The pDNA Delivery via MreB-Mediated Energy Dependent Pathway

Purple photosynthetic bacteria use far-red light energy for their growth. The pDNA delivery into *R. sulfidophilum* cells was investigated in the dark to investigate the effect of light on the delivery efficiency (Figure 5a). The efficiency was not affected in the dark, indicating that pDNA delivery is not dependent on light. In the case of plant cells, the pDNA delivery by dTat-Sar-EED was reported to be suppressed at 4 °C, inhibiting the energy-dependent process [18]. To check this observation, the pDNA delivery into *R. sulfidophilum* was examined at 4 °C. The efficiency was greatly decreased at 4 °C (Figure 5a), suggesting that pDNA delivery process was an energy-dependent process.

The delivery efficiency by dTat-Sar-EED into plant cells is shown to be significantly decreased in the presence of Wortmannin which is an inhibitor of endocytosis [18], suggesting that pDNA delivered into plant cells via endocytosis. Endocytosis is conserved process in eukaryotic cells, whereas it has not been identified in bacteria. Actin cytoskeleton plays essential role in the process of endocytosis [36,37]. MreB is the bacterial actin homologue and is involved in cell polarity and cell division [38,39]. A22 (S-(3,4-dichlorobenzyl) isothiourea) is known to be a competitive inhibitor of ATP binding by MreB [40,41]. To examine the effect of A22 on the delivery efficiency, the pDNA delivery was examined in the presence of 1, 10, 100 μg/mL of A22. The efficiency was largely inhibited by the presence of 100 μg/mL of A22 (Figure 5b). This suggests that pDNA delivery into *R. sulfidophilum* take place via MreB-mediated pathway. It is noted that cell growth was slightly inhibited in the presence of 100 μg/mL of A22 (Appendix A). A22 is involved in several cellular processes such as chromosome segregation, regulation of cell shape and cell polarity [38,40,42]. There is a possibility that decrease of the delivery efficiency might be caused by side effects of A22.

### 2.5. The dTat-Sar-EED Mediated pDNA Delivery into R. sulfidophilum

The dTat-Sar-EED5 mediated pDNA delivery into *R. sulfidophilum* cells is summarized in Figure 6. The pDNA complex is formed by electrostatic interaction between negatively charged DNA and positively charged amino acids (lysine and histidine). Because the BP100-(KH)_9_/plasmid DNA complex exhibited higher transfection efficiency compared to the (KH)_9_/plasmid DNA complex in plant cells [18], BP100 was fused to lysine and histidine for efficient pDNA delivery.

The dTat-Sar-EED5 induced cellular internalization of pDNA complex into *R. sulfidophilum* cells was successfully demonstrated in this study. Tat is shown to stimulate cellular uptake by inducing micropinocytosis, which is one of the endocytosis pathways. Macropinocytosis is an actin-mediated process and is accompanied by plasma membrane raffling and macropinosome formation [36]. Though endocytosis has not been found in bacteria, endocytosis-like uptake was observed in *Gemmata obscuriglobus* [43]. This protein uptake is shown to be an energy-dependent process and internalized proteins are associated with vesicle membranes. MreB might be involved in cellular internalization of pDNA complexes into marine purple photosynthetic bacteria. After the uptake, pDNA complex is destabilized because of the EED domain. Escape from endosome and release into the cytoplasm is a limiting step of the delivery process in eukaryotic cells. The EED domain considered to destabilize the endosomal lipid bilayer membrane by hydrophobic amino acids. Lönn et al. [17] found that the EED domain, containing two indole rings (WW, tryptophan-tryptophan amino acid residues) or one indole ring and two phenyl groups (FWF, phenylalanine-tryptophan-phenylalanine amino acid residues), enhanced cytoplasmic delivery in human cells. The delivery by dTat-Sar-EED into plant cells is shown to occur via energy-dependent endocytosis pathways based on the results of the inhibitor assays [16,18]. However, detailed cellular internalization mechanism by dTat-Sar-EED has not been clarified yet, even in plant cells. Further investigations are required to understand the mechanism in bacteria.

### 2.6. The pDNA Delivery into E. coli Cells

To explore the possibility and applicability of the peptides, namely, dTat-Sar-EED5 and BP100-(KH)_9_, the peptide-mediated pDNA delivery system into *E. coli* (DH5α) cells was examined. The plasmid pBS-ldhGFP [33] carrying ampicillin resistant gene and GFP was used for the pDNA delivery into *E. coli* cells. No resistant colonies were found when only plasmid DNA was introduced into *E. coli* cells. As shown in Figure 7a, resistant colonies were recovered by addition of pDNA complex composed of pBS-ldhGFP and BP100-(KH)_9_. Resistant colonies showed yellow-green color because of GFP expression. The pDNA delivery was confirmed by plasmid extraction (Appendix A), indicating that pDNA was successfully introduced into *E. coli* cells. Figure 7b shows the concentration effects of dTat-Sar-EED5 on the delivery efficiency. The highest efficiency was obtained without dTat-Sar-EED5, indicating that dTat-Sar-EED5 did not enhance the pDNA delivery into *E. coli* unlike *R. sulfidophilum*.

### 2.7. Mechanism of the Peptide Functions against Bacterial Cells

The expression of the antibiotics resistance gene from plasmid DNA results in antibiotics-resistant colonies. CPPs are reported to use for the delivery in some bacteria species [44,45]. In most cases, the delivery is evaluated by observation of fluorescence probe. Cell-penetrating efficiency in *E. coli* has been examined using a library of TAMRA-labeled 55 CPPs [46]. In this study, the delivery efficiency by dual peptide system was estimated by counting the colonies after antibiotic selection. Counting the colony can evaluate direct transformation efficiency, rather than only the cellular internalization. Thus, plasmid DNA was successfully delivered into *R. sulfidophilum* cells using dTat-Sar-EED5 peptide. 

However, dTat-Sar-EED5 did not enhance the pDNA delivery in *E. coli* cells (Figure 7b). Plasmid DNA alone could not be delivered into *E. coli* cells. Furthermore, the delivery efficiency was 12-fold higher at N/P ratio 5.0 than N/P ratio 2.0. These results suggest that peptide is indispensable for plasmid DNA delivery into *E. coli* cells. One possible explanation for the difference of dTat-Sar-EED5 effect between *R. sulfidophilum* and *E. coli* is the cytotoxicity of dTat-Sar-EED5. The optimum concentration of dTat-Sar-EED5 was 50–100 μM in the case of *R. sulfidophilum*, whereas the highest transformation efficiency was found at 10 μM, ranging from 1 to 100 μM dTat-Sar-EED5 in *E. coli*. This result implies that *E. coli* is more sensitive to dTat-Sar-EED5 compared to *R. sulfidophilum*. The composition of cell membrane varies among different bacterial species [47,48]. Different membrane properties between two bacterial species might lead to distinct sensitivity to dTat-Sar-EED5. The peptide-based delivery system might be applicable for other bacterial species, though optimization of delivery conditions is needed for high delivery efficiency.

## 3. Materials and Methods 

### 3.1. Culture Conditions

*R. sulfidophilum* were cultured in the Japan Collection of Microorganisms (JCM) 520-medium (https://www.jcm.riken.jp/cgi-bin/jcm/jcm_grmd?GRMD = 520) containing the following components per liter: KH_2_PO_4_, 0.5 g; CaCl_2_·2H_2_O, 0.25 g; MgSO_4_·7 H_2_O, 3.0 g; NH_4_Cl, 0.68 g; NaCl, 20 g; sodium malate, 3.0 g; sodium pyruvate, 3.0 g; yeast extract, 0.4 g; ferric citrate, 0.25 mg; vitamin B12, 2 mg; ZnCl_2_·5 H_2_O, 70 μg; MnCl_2_·4 H_2_O, 100 μg; H_3_BO_3_, 60 μg; CoCl_2_·6 H_2_O, 200 μg; CuCl_2_·2 H_2_O, 20 μg; NiCl_2_·6 H_2_O, 20 μg; and Na_2_MoO_4_·H_2_O, 40 μg. The pH was adjusted to 6.8 with NaOH. *R. sulfidophilum* were grown under continuous far-red LED light (730 nm, VBP-L24-C3, Valore, Tokyo, Japan) conditions at 30 °C in plastic tubes filled with medium to the tops of the necks. The growth of *R. sulfidophilum* was monitored by performing optical density measurements of the cells at 660 nm using a spectrophotometer. MreB Perturbing Compound A22 (carbamimidothioic acid, (3,4-dichlorophenyl)methyl ester, monohydrochloride) was purchased from Cayman Chemical (Ann Arbor, MI, USA) and dissolved in DMSO as 10 mg/mL stock solution. Different concentrations of A22 (1, 10 and 100 μg/mL) were added to the culture to evaluate its inhibitory effect on plasmid DNA delivery.

*E. coli* DH5α cells were cultured in LB broth and LB agar (Becton Dickinson and Company, Franklin Lakes, NJ, USA) at 37 °C. 100 mg/L ampicillin was used for selection. 

### 3.2. Preparation of Peptide-Plasmid DNA Complex 

The pBBR1MCS-2 [33] and pBS-ldhGFP (Addgene plasmid #27170 [49]) were prepared using QIAprep Spin Miniprep kit (QIAGEN, Dusseldorf, Germany) according to the standard protocol. The peptides used in this study including dTat-Sar-EED5: d(RRRQRRKKR)-(Sar)_6_-GFWFG and BP100-(KH)_9_: KKLFKKILKYLKHKHKHKHKHKHKHKHKH were obtained from the Support Unit for Bio-Material Analysis in RIKEN Center for Brain Science (Wako, Japan). 1 μg of plasmid DNA was mixed with peptides BP100-(KH)_9_ at indicated N/P ratio (0.1, 0.5, 1, 2, 5 and 10; stock solution is 1 mg/mL) and incubated at room temperature for 30 min to form pDNA complexes. 

### 3.3. Transformation of R. sulfidophilum and E. coli Cells 

The log phase cultures of *R. sulfidophilum* (1.5 mL) were washed and suspended with water. Washed cells were mixed with dTat-Sar-EED5 peptides at indicated concentrations (1, 10, 50, 100, 200, 300 and 500 μM; stock solution was 2 mg/mL) and incubated for 30 min at 30 °C under far-red light illumination. The pDNA complex (BP100-(KH)_9_/pBBR1MCS-2) was mixed with dTat-Sar-EED5 treated cells and incubated for 1 h. The reaction solutions were removed by centrifugation and then suspended with growth medium without kanamycin and cultured at 30 °C overnight under far-red illumination. Next day, cell cultures were spread in agar plates containing 100 mg/L kanamycin and cultured at 30 °C under the far-red light.

1.5 mL of log phase cultures of *E. coli* DH5α were washed and suspended with water. Washed cells were mixed with dTat-Sar-EED5 peptides at indicated concentrations (1, 10, 50, 100, 200, 300 and 500 μM; stock solution is 2 mg/mL) and incubated for 10 min at 37 °C. The pDNA complex (BP100-(KH)_9_/pBS-ldhGFP) was mixed with dTat-Sar-EED5 treated cells and incubated for 30 min. The reaction solutions were removed by centrifugation and then suspended with LB broth without ampicillin and incubated at 37 °C for 1.5 h. Then, cell cultures were spread in LB plates containing 100 mg/L ampicillin and cultured at 37 °C overnight.

### 3.4. Evans Blue Staining

*R. sulfidophilum* cells were incubated with various concentrations of dTat-Sar-EED5 at 30 °C for 1 h. Cell viability was determined by incubating the cells with 1.5% of Evans blue (Sigma-Aldrich, MO, USA) in distilled water, followed by solubilization of bound stain in 50% aqueous methanol containing 1% SDS and spectrophotometric quantification at 600 nm.

### 3.5. Formation and Stability of pDNA Complexes

Electrophoretic mobility shift assays were performed to detect the stabilities of complexes formed between the BP100-(KH)_9_ peptide and pDNA [12]. BP100-(KH)_9_ was added to plasmid DNA (1.0 μg) at various N/P ratios (0, 0.1, 0.5, 1, 2, 5 and 10), adjusted to a final volume of 20 μL, and electrophoresed on a 1% agarose gel for 30 min at 100 V.

### 3.6. Dynamic Light Scattering (DLS) Analysis 

BP100-(KH)_9_ peptide was mixed with pDNA at various N/P ratios (0.1, 0.5, 1, 2, 5 and 10) and adjusted to a final volume of 100 μL (Size and PGI) and 800 μL (zeta potential) with autoclaved Milli-Q water or growth medium. The solutions were thoroughly mixed by repeated pipetting and allowed to stabilize for 30 min at room temperature. DLS and zeta potential analyses were carried out using a zeta potentiometer (Zetasizer Nano-ZS; Malvern Instruments, Ltd., Worcestershire, UK).

## 4. Conclusions

In this study, a dual peptide system composed of dTat-Sar-EED5 and BP100-(KH)_9_ peptides enabled efficient pDNA uptake and integration into *R. sulfidophilum* cells. This pDNA delivery was an actin homolog MreB-mediated energy-dependent process. Stable formation of the pDNA complex, long recovery period and actively proliferating cells are important factors for efficient delivery into *R. sulfidophilum* cells. Additionally, this peptide-based pDNA delivery was also applicable to *E. coli* cells though dTat-Sar-EED peptide did not enhance the efficiency. 

CPPs originally used to facilitate biomolecule delivery in mammalian cells. Recently, peptide-derived technology has been applied for delivery of biomolecules in plant cells. The peptide-based system was also able to deliver pDNA into bacterial cells, as was successfully demonstrated in this study. The peptide-mediated gene transfer method developed in this study is easy and simple, because preparation of competent cells and special plasmids is not required. This method can extend the possibilities of microbial biotechnology in future. 

## Figures and Tables

**Figure 1 ijms-21-08625-f001:**
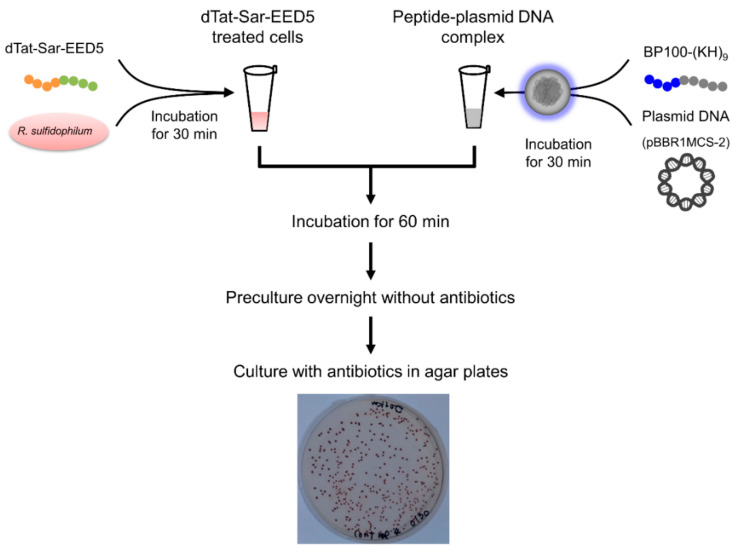
Procedure for the plasmid DNA delivery in this study. Cultures of *R. sulfidophilum* were washed and suspended with water. Washed cells were mixed with dTat-Sar-EED5 and incubated for 30 min. The plasmid DNA was mixed with BP100-(KH)_9_ peptide and incubated at room temperature for 30 min to form pDNA complexes. The pDNA complex was mixed with dTat-Sar-EED5 treated cells and incubated for 60 min. Cells were cultured overnight without kanamycin. On the next day, cell cultures were spread in agar plates containing kanamycin and cultured at 30 °C under the far-red light.

**Figure 2 ijms-21-08625-f002:**
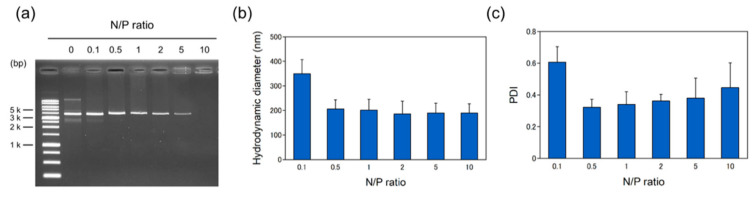
Characterization of pDNA complexes. (**a**) The electrical stability of the pDNA complexes. (**b**) Size of the pDNA complexes. (**c**) PDI of the pDNA complexes. Plasmid DNA (pBBR1MCS-2) was mixed with BP100-(KH)_9_ at different N/P ratio, with a fixed amount of pDNA (1 μg). Data are the mean ± SD of at least three experiments.

**Figure 3 ijms-21-08625-f003:**
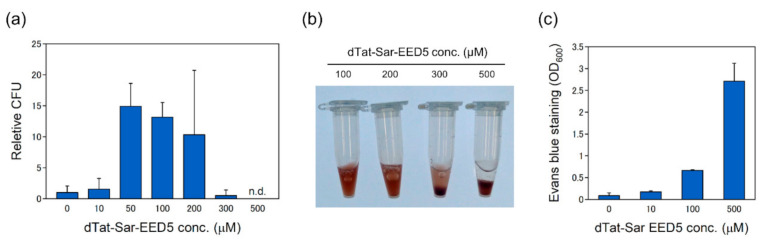
Effects of dTat-Sar-EED5 on *R. sulfidophilum* cells. (**a**) The pDNA delivery efficiency at different concentrations of dTat-Sar-EED5. The data were normalized by ‘0 μM dTat-Sar-EED5′. n.d., not detected. (**b**) Effects of dTat-Sar-EED5 on *R. sulfidophilum* cell cultures. Photos were taken just after the dTat-Sar-EED5 treatment. (**c**) Evans blue staining of *R. sulfidophilum* cells after the dTat-Sar-EED5 treatment. Data are the mean ± SD of at least three cultures.

**Figure 4 ijms-21-08625-f004:**
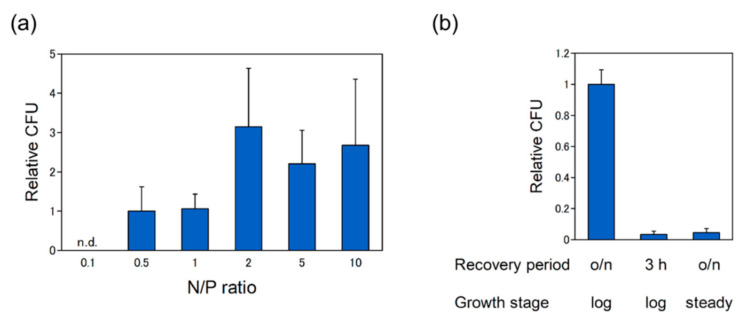
Optimization for the pDNA delivery into *R. sulfidophilum* cells. (**a**) Effect of N/P ratio on the efficiency. The pDNA complexes was prepared using a fixed amount of pDNA (1 μg) at N/P ratio 0.1, 0.5, 1, 2, 5 and 10. The data were normalized by ‘N/P ratio 0.5′. n.d., not detected. (**b**) Effects of recovery period and cell growth stage on the delivery efficiency. The log phase (OD_660_ of around 1.0) cells and the stationary phase (OD_660_ of around 4.0) cells were used for experiments. The concentration of dTat-Sar-EED5 was fixed to 100 μM. Data are the mean ± SD of at least three cultures. The data were normalized by ‘o/n recovery period and log phase cells’ condition.

**Figure 5 ijms-21-08625-f005:**
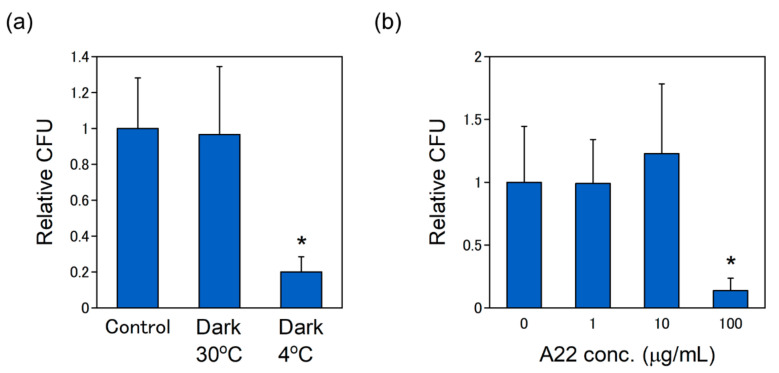
The pDNA delivery efficiency under various conditions. (**a**) Effect of growth light and temperature conditions on the transformation efficiency. The pDNA delivery was conducted at 30 °C or 4 °C under far-red illumination or in the dark. The data were normalized by the ‘control’ condition (far-red light at 30 °C). Asterisk shows the significant difference compared to the control (*p* < 0.01). (**b**) Effect of A22 on the delivery efficiency. The pDNA delivery was carried out in the presence of 0, 1, 10 and 100 μg/mL of A22. The concentration of dTat-Sar-EED5 was fixed to 100 μM. The data were normalized by ‘0 μg/mL A22′. Data are the mean ± SD of four cell cultures. Asterisk shows the significant difference compared to the control (without A22) (*p* < 0.01). Statistically significant differences between samples were determined by the Student *t*-test.

**Figure 6 ijms-21-08625-f006:**
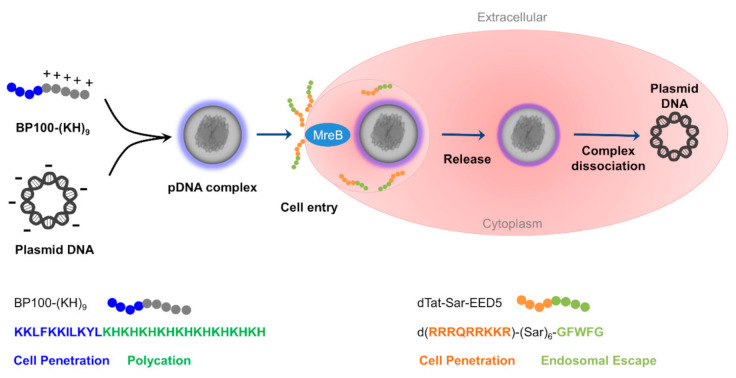
Schematic representation of the dual peptide delivery system of plasmid DNA.

**Figure 7 ijms-21-08625-f007:**
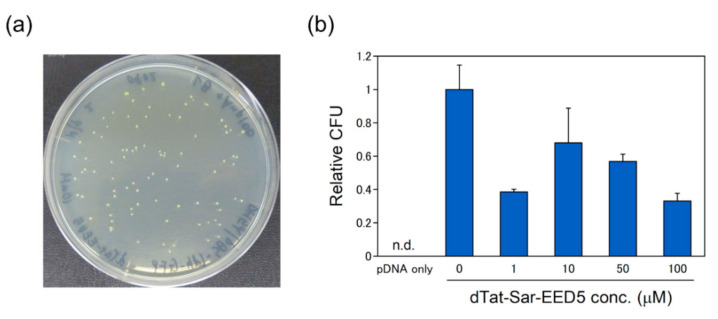
The pDNA delivery into *E. coli* cells. (**a**) Ampicillin-resistant colonies of *E. coli*. The pDNA complex (BP100-(KH)_9_/pBS-ldhGFP) at N/P ratio 2.0 was introduced into 10 μM dTat-Sar-EED5 treated *E. coli* cells. (**b**) Effects of dTat-Sar-EED5 concentrations on the transformation efficiency of *E. coli*. The pDNA delivery was carried out under different concentrations of dTat-Sar-EED5 (0, 1, 10, 50 and 100 μM). The N/P ratio was set to 5.0, where a fixed amount of plasmid DNA (1 μg) was mixed with various amounts of peptide BP100-(KH)_9_. ‘pDNA only’ means plasmid DNA alone without BP100-(KH)_9_ and dTat-Sar-EED5. ‘0 µM of dTat-Sar-EED5′ means pDNA complex (BP100-(KH)_9_/plasmid DNA) without dTat-Sar-EED5. Data were normalized by ‘0 μM dTat-Sar-EED5′. Data are the mean ± SD of three cell cultures. n.d., not detected.

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
