# Peer review of "Peptide-Mediated Gene Transfer into Marine Purple Photosynthetic Bacteria"

_ijms, 2020, doi:10.3390/ijms21228625_

Round 1
Reviewer 1 Report
The manuscript entitled "Peptide-mediated gene transfer into marine purple photosynthetic bacteria" sets out to investigate the dual peptide-based delivery system into marine purple photosynthetic bacteria and E. coli. The authors reported that the dual peptide system [dTat-Sar-EED5 and BP100-(KH)9] enhanced DNA uptake and integration into Rhodovulum sulfidophilum cells but not in E. coli cells. The study is on a topic of relevance and general interest to the readers of the journal. I found the paper to be overall well written and felt confident that the authors performed a careful and thorough experiment and spectral processing. However, I have several concerns about the presentation of the data that should be addressed before publication.
- The authors are highly recommended to avoid using a personal pronoun (e.g., We, our, etc.); they can use the third party in the past tense's passive voice.
- Keywords: I suggest the authors use different keywords that are not included in the title
- The authors need carefully to read through the manuscript to correct typos and grammars to improve the manuscript. Below are a few examples.
- In the abstract, the author should focus on the current study, and don't report a result from a previous study (Line 15-17), move this statement to the introduction or the discussion section.
- Figure 2a: indicate the band size on the gel
- Page 4 lines120-129: this paragraph is more fit in the material and method section, not in the results section
- Page 4 line 132, what enzyme used and what are the expected bands
- Page 6, line 205: figures should present in order in the text, so show the figures in order (figure 6 should be before figure 7) or rearrange the figure number accordingly.
- Page 6, line 221: add the reference number after Lonn et al.
- Page 6, line 221, what WW stand for?
- Page 7 line 232: as I said, figure 6 should be presented before figure 7 or change the figure order
- Figure 6b: could the authors explain the difference between pDNA only and 0 µM of dTat-Sar-EED5 concentration, this very confusing. pDNA only and 0 µM of dTat-Sar-EED5 should be the same, but why giving completely different results?
- Page 8, line 269: What JCM stand for?
- Page 8, line 278: What is the concentration of A22?
- Page 8, line 288: the authors need to indicate the N/P ratio used herein in the materials and methods section
- Page 8, line 292: the authors need to indicate the dTat-Sar-EED5 concentration here
- Page 8, line 293: the pDNA complex, what DNA plasmid used to form this complex
- Page 9, line 299: the authors need to indicate the dTat-Sar-EED5 concentration here
- Page 9, line 300: the pDNA complex, what DNA plasmid used to form this complex
- Page 9, line 306: what is the source of Evans blue
- Page 9, line 310-313: this section needs a reference and also be specific with the N/P ratio that used here and the same for line 315, be specific about the ratio
- Supplementary material, add the figure captions in the supplementary material file as well and indicate the band size on the gel
Author Response
Reviewer #1:
Comment 1: The manuscript entitled "Peptide-mediated gene transfer into marine purple photosynthetic bacteria" sets out to investigate the dual peptide-based delivery system into marine purple photosynthetic bacteria and E. coli. The authors reported that the dual peptide system [dTat-Sar-EED5 and BP100-(KH)9] enhanced DNA uptake and integration into Rhodovulum sulfidophilum cells but not in E. coli cells. The study is on a topic of relevance and general interest to the readers of the journal. I found the paper to be overall well written and felt confident that the authors performed a careful and thorough experiment and spectral processing. However, I have several concerns about the presentation of the data that should be addressed before publication.
Response: Thank you for the encouraging comments and compliments.
Comment 2: The authors are highly recommended to avoid using a personal pronoun (e.g., We, our, etc.); they can use the third party in the past tense's passive voice.
Response: We have revised the sentences accordingly to remove personal pronoun.
- Along with the comment, we revised the text shown as below.
“Original”
“Modified”
- (page 1, line 13)
“We have focused on marine purple photosynthetic bacteria as microbial production hosts.”
“Marine purple photosynthetic bacteria are one of the research focuses as microbial production hosts.”
- (page 1, line 16)
“In this study, we combined this complex with Tat-derived peptide (dTat-Sar-EED) containing D-amino acids of Tat and endosomal escape domain (EED) connected by sarcosine linkers for plasmid DNA delivery into Rhodovulum sulfidophilum, a marine purple photosynthetic bacterium.”
“In this study, for the first time, a dual peptide-based transformation method, which combined cell penetrating peptide (CPP), cationic peptide and Tat-derived peptide (dTat-Sar-EED) (containing D-amino acids of Tat and endosomal escape domain (EED) connected by sarcosine linkers) had successfully delivered plasmid DNA into Rhodovulum sulfidophilum, a marine purple photosynthetic bacterium.”
- (page 1, line 24)
“We also demonstrated that peptide-mediated delivery was also applicable for E. coli cells. Thus, the novel peptide-based transformation method is applicable for a wide range of bacteria.”
“Besides, this peptide-mediated delivery method was also applicable for E. coli cells. Thus, a wide range of bacteria could be genetically transformed by using this novel dual peptide-based transformation method.”
- (page 2, line 52)
“We designed carrier peptides composed of cell penetrating peptides (CPPs) and polycation amino acids (lysine and histidine) and used for bimolecular delivery into plant cells.”
“Carrier peptides were designed to compose cell penetrating peptides (CPPs) and polycation amino acids (lysine and histidine) and used for bimolecular delivery into plant cells.”
- (page 2, line 55)
“Our peptide-based carrier can deliver various types of molecules such as DNA [12], RNA [13] and even protein into plant cells [14,15].”
“These peptide-based carriers could deliver various types of molecules such as DNA [12], RNA [13] and even protein into plant cells [14,15].”
- (page 2, line 57)
“Recently, we reported novel carrier peptides, dTat-Sar-EED4 and dTat-Sar-EED5 [16].”
“Recently, novel carrier peptides, dTat-Sar-EED4 and dTat-Sar-EED5 were reported [16].”
- (page 2, line 61)
“We also found that combination of peptide-plasmid DNA (pDNA) complex and dTat-Sar-EED peptide enhanced the delivery efficiency into plant cells [18].”
“Interestingly, a combination of peptide-plasmid DNA (pDNA) complex and dTat-Sar-EED peptide enhanced the delivery efficiency into plant cells [18].”
- (page 2, line 71)
“We reported that several species of marine purple photosynthetic bacteria could produce polyhydroxyalkanoate (PHA) biopolyesters [25-27].”
“Several species of marine purple photosynthetic bacteria that could produce polyhydroxyalkanoate (PHA) biopolyesters had been reported [25-27].”
- (page 2, line 77)
“Recently, we succeeded the preparation of chemically competent cells for marine purple photosynthetic bacteria by calcium chloride treatment [32].”
“Recently, the preparation of chemically competent cells for marine purple photosynthetic bacteria by calcium chloride treatment was succeeded [32].”
- (page 2, line 80)
“Although natural transformation is another way of bacterial transformation, it has not been established in marine purple photosynthetic bacteria as far as we know.”
“Although natural transformation is another way of bacterial transformation, to the best of knowledge, it has not been established in marine purple photosynthetic bacteria.”
- (page 2, line 81)
“In this study, we examined the dual peptide-based delivery system into marine purple photosynthetic bacteria.”
“In this study, the dual peptide-based delivery system into marine purple photosynthetic bacteria was examined.”
- (page 2, line 94)
“Therefore, we used BP100-(KH)9 for the delivery into marine purple photosynthetic bacteria.”
“Therefore, BP100-(KH)9 was used for the delivery into marine purple photosynthetic bacteria.”
- (page 5, line 162)
“Next, we examined effects of the recovery duration and cell growth stage on the delivery efficiency (Fig. 4b).”
“Next, effects of the recovery duration and cell growth stage on the delivery efficiency were examined (Fig. 4b).”
- (page 6, line 218)
“We demonstrated that the dTat-Sar-EED5 induced cellular internalization of pDNA complex into R. sulfidophilum cells …”
“The dTat-Sar-EED5 induced cellular internalization of pDNA complex into R. sulfidophilum cells was successfully demonstrated in this study.”
- (page 7, line 240)
“To explore the possibility and applicability of the peptides, namely, dTat-Sar-EED5 and BP100-(KH)9, we examined the peptide-mediated pDNA delivery system into E. coli (DH5α) cells.”
“To explore the possibility and applicability of the peptides, namely, dTat-Sar-EED5 and BP100-(KH)9, the peptide-mediated pDNA delivery system into E. coli (DH5α) cells was examined.”
- (page 7, line 253)
“We also examined cell-penetrating efficiency in E. coli using a library of TAMRA-labeled 55 CPPs [46].”
“Cell-penetrating efficiency in E. coli had been examined using a library of TAMRA-labeled 55 CPPs [46].”
- (page 7, line 256)
“Thus, we concluded that plasmid DNA successfully delivered into R. sulfidophilum cells using dTat-Sar-EED5 peptide.”
“Thus, plasmid DNA had been successfully delivered into R. sulfidophilum cells using dTat-Sar-EED5 peptide.”
- (page 7, line 259)
“We found that dTat-Sar-EED5 did not enhance the pDNA delivery in E. coli cells (Fig. 6b).”
“However, dTat-Sar-EED5 did not enhance the pDNA delivery in E. coli cells (Fig. 7b).”
- (page 7, line 260)
“Furthermore, we found that the delivery efficiency was 12-fold higher at N/P ratio 5.0 than N/P ratio 2.0.”
“Furthermore, the delivery efficiency was 12-fold higher at N/P ratio 5.0 than N/P ratio 2.0.”
- (page 9, line 341)
“In this study, we demonstrated that dual peptide system composed of dTat-Sar-EED5 and BP100-(KH)9 peptides enabled efficient pDNA uptake and integration into R. sulfidophilum cells.”
“In this study, dual peptide system composed of dTat-Sar-EED5 and BP100-(KH)9 peptides had enabled efficient pDNA uptake and integration into R. sulfidophilum cells.”
- (page 9, line 345)
“We also demonstrated that peptide-based pDNA delivery was applicable to E. coli cells though dTat-Sar-EED peptide did not enhance the efficiency.”
“Besides, this peptide-based pDNA delivery was also applicable to E. coli cells though dTat-Sar-EED peptide did not enhance the efficiency.”
- (page 9, line 348)
“We demonstrated that peptide-based system also could deliver pDNA into bacterial cells.”
“Peptide-based system also could deliver pDNA into bacterial cells, which was successfully demonstrated in this study.”
- (page 9, line 351)
“We hope this method can extend the possibilities of microbial biotechnology in future.”
“This method can extend the possibilities of microbial biotechnology in future.”
Comment 3: Keywords: I suggest the authors use different keywords that are not included in the title
Response: We have revised “marine purple photosynthetic bacteria” to “Rhodovulum sulfidophilum”.
- Along with the comment, we revised the keyword. (page 1, line 27).
“Keywords: transformation; plasmid DNA delivery; Rhodovulum sulfidophilum; cell penetrating peptide”
Comment 4: In the abstract, the author should focus on the current study, and don't report a result from a previous study (Line 15-17), move this statement to the introduction or the discussion section.
Response: We have removed this statement from the abstract and modified the sentences.
- Along with the comment, we revised the text. (page 1, line 16-20).
“…transformation of marine purple photosynthetic bacteria so far. We developed peptide-based delivery methods into plant cells using ionic complex of fusion peptides consisting of a cell penetrating peptide (CPP) and cationic peptides. In this study, we combined this complex with Tat-derived peptide (dTat-Sar-EED) containing D-amino acids of Tat and endosomal escape domain (EED) connected by sarcosine linkers for plasmid DNA delivery into Rhodovulum sulfidophilum, a marine purple photosynthetic bacterium.”
“…transformation of marine purple photosynthetic bacteria so far. In this study, for the first time, a dual peptide-based transformation method, which combined cell penetrating peptide (CPP), cationic peptide and Tat-derived peptide (dTat-Sar-EED) (containing D-amino acids of Tat and endosomal escape domain (EED) connected by sarcosine linkers) had successfully delivered plasmid DNA into Rhodovulum sulfidophilum, a marine purple photosynthetic bacterium.”
Comment 5: Figure 2a: indicate the band size on the gel
Response: We have added the band sizes (DNA marker) on the gel.
- Along with the comment, we revised the Figure 2a.
Figure 2. Characterization of pDNA complexes. (a) The electrical stability of the pDNA complexes.
Comment 6: Page 4 lines120-129: this paragraph is more fit in the material and method section, not in the results section.
Response: Yes, we partially agreed with this point. We have removed a few sentences that are more appropriate in the material and method section. However, it is necessary to explain and discuss the rationale for some of these steps.
- Along with the comment, we revised the text in this paragraph. (page 4, line 121-127).
“The R. sulfidophilum cell cultures (1.5 mL) were washed and resuspend in water before added with dTat-Sar-EED5 (Fig. 1). Substitution with water from growth medium was important because growth medium contained high concentrations of NaCl (2%) leading to disturbance of ionic pDNA complex formation. The pDNA complex showed bimodal distribution in growth medium and average size was quite large (916 ± 40 nm). In addition, the pDNA complex in growth medium exhibited negative zeta potential (-18.9 ± 1.6 mV) which reported low gene delivery efficiency in plant cells [12].”
Comment 7: Page 4 line 132, what enzyme used and what are the expected bands
Response: BglII and HindIII restriction enzymes were used in this experiment (Figure S1).
- Along with the comment, we revised the text. (page 4, line 130)
“The presence of pDNA from kanamycin resistant colonies was verified by pDNA extraction followed by restriction digestion with BglII and HindIII (Fig. S1), …”
- Along with the comment, we revised the figure caption of Figure S1
“Figure S1. Plasmid extraction and restriction digestion of kanamycin resistant colonies of R. sulfidophilum. Plasmid solutions extracted from kanamycin resistant colonies were digested with BglII and HindIII and analyzed by …”
Comment 8: Page 6, line 205: figures should present in order in the text, so show the figures in order (figure 6 should be before figure 7) or rearrange the figure number accordingly. Response:
- Along with the comment, we re-arranged the order for Figure 6 and Figure 7, and also figure numbers accordingly.
Figure 6. Schematic representation of the dual peptide delivery system of plasmid DNA.
Figure 7. The pDNA delivery into E. coli cells.
- In the text. (page 6, line 213)
“… pDNA delivery into R. sulfidophilum cells is summarized in Fig. 6.”
- In the text. (page 7, line 244)
“As shown in Fig. 7a, resistant colonies were recovered by addition of pDNA complex…”
- In the text. (page 7, line 247)
“…pDNA was successfully introduced into E. coli cells. Fig. 7b shows the concentration…”
- In the text. (page 7, line 259)
“…dTat-Sar-EED5 did not enhance the pDNA delivery in E. coli cells (Fig. 7b).”
Comment 9: Page 6, line 221: add the reference number after Lonn et al.
Response: A reference number is added after Lonn et al.
- Along with the comment, we revised the text. (page 6, line 228)
“Lönn et al. [17] found that EED domain containing two indole rings …”
Comment 10: Page 6, line 221, what WW stand for?
Response: In this context, ‘W’ is an abbreviation for tryptophan amino acid residue.
- Along with the comment, we revised the text (page 6, line 229)
“… EED domain containing two indole rings (WW, tryptophan-tryptophan amino acid residues) or one indole ring and two phenyl groups (FWF, phenylalanine-tryptophan-phenylalanine amino acid residues) enhanced cytoplasmic delivery in human cells.”
Comment 11: Page 7 line 232: as I said, figure 6 should be presented before figure 7 or change the figure order.
Response:
- Along with the comment, we re-arranged the order for Figure 6 and Figure 7, and also figure numbers accordingly.
Figure 6. Schematic representation of the dual peptide delivery system of plasmid DNA.
Figure 7. The pDNA delivery into E. coli cells.
- In the text. (page 6, line 213)
“… pDNA delivery into R. sulfidophilum cells is summarized in Fig. 6.”
- In the text. (page 7, line 244)
“As shown in Fig. 7a, resistant colonies were recovered by addition of pDNA complex…”
- In the text. (page 7, line 247)
“…pDNA was successfully introduced into E. coli cells. Fig. 7b shows the concentration…”
- In the text. (page 7, line 259)
“…dTat-Sar-EED5 did not enhance the pDNA delivery in E. coli cells (Fig. 7b).”
Comment 12: Figure 7b: could the authors explain the difference between pDNA only and 0 µM of dTat-Sar-EED5 concentration, this very confusing. pDNA only and 0 µM of dTat-Sar-EED5 should be the same, but why giving completely different results?
Response: We apologize for making the confusion here.
‘pDNA only’ and ‘0 µM of dTat-Sar-EED5’ are different conditions. ‘pDNA only’ means plasmid DNA alone was introduced into cells without BP100-(KH)9 and dTat-Sar-EED5. While ‘0 µM of dTat-Sar-EED5’ indicates pDNA complex (BP100-(KH)9/plasmid DNA) was introduced into cells without dTat-Sar-EED5.
We have added these definitions in the figure caption.
- Along with the comment, we revised the figure caption of Figure 7.
Figure 7. The pDNA delivery into E. coli cells. (a) Ampicillin resistant colonies of E. coli. The pDNA complex (BP100-(KH)9/pBS-ldhGFP) at N/P ratio 2.0 was introduced into 10 μM dTat-Sar-EED5 treated E. coli cells. (b) Effects of dTat-Sar-EED5 concentrations on the transformation efficiency of E. coli. The pDNA delivery was carried out under different concentrations of dTat-Sar-EED5 (0, 1, 10, 50 and 100 μM). The N/P ratio was set to 5.0, where a fixed amount of plasmid DNA (1 μg) was mixed with various amounts of peptide BP100-(KH)9. ‘pDNA only’ means plasmid DNA alone without BP100-(KH)9 and dTat-Sar-EED5. ‘0 µM of dTat-Sar-EED5’ means pDNA complex (BP100-(KH)9/plasmid DNA) without dTat-Sar-EED5. Data are the mean ± SD of three cell cultures.
Comment 13: Page 8, line 269: What JCM stand for?
Response: In this context, JCM is an abbreviation for Japan Collection of Microorganisms
- Along with the comment, we revised the text. (page 8, line 284)
“R. sulfidophilum were cultured in the Japan Collection of Microorganisms (JCM) 520-medium …”
Comment 14: Page 8, line 278: What is the concentration of A22?
Response: Stock concentration and working concentrations of A22 are added into the text.
- Along with the comment, we revised the text. (page 8, line 293)
“MreB Perturbing Compound A22 (carbamimidothioic acid, (3,4-dichlorophenyl)methyl ester, monohydrochloride) was purchased from Cayman Chemical (Ann Arbor, MI, USA) and dissolved in DMSO as 10 mg/mL stock solution. Different concentrations of A22 (1, 10 and 100 μg/mL) was added to the culture to evaluate its inhibitory effect on plasmid DNA delivery.”
Comment 15: Page 8, line 288: the authors need to indicate the N/P ratio used herein in the materials and methods section
Response: Stock concentration of BP100-(KH)9 and N/P ratio are added into the text.
- Along with the comment, we revised the text. (page 9, line 306)
“1 μg of plasmid DNA was mixed with peptide BP100-(KH)9 at indicated N/P ratio (0.1, 0.5, 1, 2, 5 and 10; stock solution is 1 mg/mL) and incubated …”
Comment 16: Page 8, line 292: the authors need to indicate the dTat-Sar-EED5 concentration here
Response: Stock concentration and working concentrations of dTat-Sar-EED5 are added into the text.
- Along with the comment, we revised the text. (page 9, line 310)
“Washed cells were mixed with dTat-Sar-EED5 peptide at indicated concentrations (1, 10, 50, 100, 200, 300 and 500 μM; stock solution is 2 mg/mL) and incubated …”
Comment 17: Page 8, line 293: the pDNA complex, what DNA plasmid used to form this complex
Response: In this context, the pDNA complex is referring to “BP100-(KH)9/pBBR1MCS-2”.
- Along with the comment, we revised the text. (page 9, line 312)
“The pDNA complex (BP100-(KH)9/pBBR1MCS-2) was mixed with dTat-Sar-EED5 …”
Comment 18: Page 9, line 299: the authors need to indicate the dTat-Sar-EED5 concentration here
Response: Stock concentration and working concentrations of dTat-Sar-EED5 are added into the text.
- Along with the comment, we revised the text. (page 9, line 318)
“cells were mixed with dTat-Sar-EED5 peptide at indicated concentrations (1, 10, 50, 100, 200, 300 and 500 μM; stock solution is 2 mg/mL) and incubated …”
Comment 19: Page 9, line 300: the pDNA complex, what DNA plasmid used to form this complex.
Response: In this context, the pDNA complex is referring to “BP100-(KH)9/pBS-ldhGFP”.
- Along with the comment, we revised the text. (page 9, line 319)
“The pDNA complex (BP100-(KH)9/pBS-ldhGFP) was mixed with dTat-Sar-EED5 …”
Comment 20: Page 9, line 306: what is the source of Evans blue
Response: The source of Evans blue is Sigma-Aldrich, MO, USA.
- Along with the comment, we revised the text. (page 9, line 326)
“Cell viability was determined by incubating the cells with 1.5% of Evans blue (Sigma-Aldrich, MO, USA) in distilled water, …”
Comment 21: Page 9, line 310-313: this section needs a reference and also be specific with the N/P ratio that used here and the same for line 315, be specific about the ratio.
Response:
- Along with the comment, we added a reference to this section. (page 9, line 330)
“Electrophoretic mobility shift assays were performed to detect the stabilities of complexes formed between the BP100-(KH)9 peptide and pDNA [12].”
Lakshmanan, M.; Kodama, Y.; Yoshizumi, T.; Sudesh, K.; Numata, K. Rapid and efficient gene delivery into plant cells using designed peptide carriers. Biomacromolecules 2013, 14, 10-16, doi:10.1021/bm301275g.
- Along with the comment, we revised the text. (page 9, line 332)
“BP100-(KH)9 was added to plasmid DNA (1.0 μg) at various N/P ratios (0, 0.1, 0.5, 1, 2, 5 and 10), adjusted…”
- Along with the comment, we revised the text. (page 9, line 335)
“BP100-(KH)9 peptide was mixed with pDNA at various N/P ratios (0.1, 0.5, 1, 2, 5 and 10) and adjusted…”
Comment 22: Supplementary material, add the figure captions in the supplementary material file as well and indicate the band size on the gel.
Response:
- Along with the comment, we added figure captions (Figure S1, S2 and S3) in the revised supplementary material file.
- Along with the comment, we added band sizes (DNA marker) on the gel in Fig. S1 and Fig. S3.
Figure S1. Plasmid extraction and restriction digestion of kanamycin resistant colonies of R. sulfidophilum.
Figure S3. Plasmid extraction from ampicillin resistant colonies of E. coli.

Reviewer 2 Report
The manuscript: Peptide-mediated gene transfer into marine purple photosynthetic bacteria describes novel method to introduce pDNA into bacteria Rhodovulum sulfidophilum using dual peptide based gene delivery system. This is a nice example of method development of biotechnological tools and potentially enables manipulation of broader range of microbes.
I found the text very well written and quite easy to follow. However, there some minor issues I want to point out.
I am a bit confused about relative CFU in most of the figures. Relative to what? It is not consistent, for instance Fig. 4 B seems to compare to the o/n preculture growth stage log (which is 1) whereas Fig. 4 A to N/P ratio one (also 1)? And how it is calculated in figure 5 B then? Could you clarify these?
How is statistical significance calculated in figure 5?
It would be also quite essential to know the general level of colonies acquired in the most efficient gene transfers (per µg of plasmid or something) i.e. the delivery efficiency. I don’t see that mentioned anywhere.
Also, the amount of DNA used in all in the experiments could be mentioned already in the Results and dicussion section or at least that it was always fixed amount. This was something I did not understand immediately from the description of the experiments.
In figure 4 B using the word Preculture is confusing since in the text it is recovery period, if I understood correctly. This could be corrected.
Here are some specific comments about the language but they are only suggestions since I am not native English speaker:
row 253: Only plasmid DNA could not.. à Plasmid DNA alone could not..
row 54: CPPs have the the ability that translocate.. à CPPs have the the ability to translocate..
row 211: I would remove the end of the sentence: and the delivery is mediated by actin homolog MreB. This is because definite proof of this is not shown. Also, in other statements thorough out the text this is taken into account.
All in all the study is very interesting in terms of genetic manipulation of microbes and the method itself could be one avenue to explore further in bacteria where gene delivery is challenging. Thus, I recommend publication after minor modifications.
Author Response
Reviewer #2:
Comment 1: The manuscript: Peptide-mediated gene transfer into marine purple photosynthetic bacteria describes novel method to introduce pDNA into bacteria Rhodovulum sulfidophilum using dual peptide based gene delivery system. This is a nice example of method development of biotechnological tools and potentially enables manipulation of broader range of microbes.
I found the text very well written and quite easy to follow. However, there some minor issues I want to point out.
Response: Thank you for the encouraging comments and compliments.
Comment 2: I am a bit confused about relative CFU in most of the figures. Relative to what? It is not consistent, for instance Fig. 4 B seems to compare to the o/n preculture growth stage log (which is 1) whereas Fig. 4 A to N/P ratio one (also 1)? And how it is calculated in figure 5 B then? Could you clarify these?
Response: We apologize for making the confusion about the relative CFU.
Actually, we made a mistake in Fig. 5b. The original Fig. 5b showed a real CFU and not a relative CFU. Therefore, we revised the Fig. 5.
Relative CFU were calculated as shown below. We have added these statements in the figure captions.
- 3a was normalized by ‘0 µM dTat-Sar-EED5’. (This means that 0 µM dTat-Sar-EED5 was 1.0)
- 4a was normalized by ‘N/P ratio at 0.5’.
- 4b was normalized by first condition (o/n recovery period and log phase cells)
- 5a was normalized by ‘control condition’ (Far-red light at 30oC)
- 5b was normalized by ‘0 µg/mL A22’.
- 7b was normalized by ‘0 µM dTat-Sar-EED5’.
- Along with the comment, we revised the figure caption of Figure 3a.
“Figure 3. Effects of dTat-Sar-EED5 on R. sulfidophilum cells. (a) The pDNA delivery efficiency at different concentrations of dTat-Sar-EED5. The data were normalized by ‘0 µM dTat-Sar-EED5’. (b) Effects of …”
- Along with the comment, we revised the figure caption of Figure 4.
Figure 4. Optimization for the pDNA delivery into R. sulfidophilum cells. (a) Effect of N/P ratio on the efficiency. The pDNA complexes was prepared using a fixed amount of pDNA (1 μg) at N/P ratio 0.1, 0.5, 1, 2, 5 and 10. The data were normalized by ‘N/P ratio 0.5’. (b) Effects of recovery period and cell growth stage on the delivery efficiency. The log phase (OD660 of around 1.0) cells and the stationary phase (OD660 of around 4.0) cells were used for experiments. The concentration of dTat-Sar-EED5 was fixed to 100 μM. Data are the mean ± SD of at least three cultures. The data were normalized by ‘o/n recovery period and log phase cells’ condition.
- Along with the comment, we revised the figure caption of Figure 5.
Figure 5. The pDNA delivery efficiency under various conditions. (a) Effect of growth light and temperature conditions on the transformation efficiency. The pDNA delivery was conducted at 30oC or 4oC under far-red illumination or in the dark. The data were normalized by ‘control’ condition (Far-red light at 30oC. Asterisk shows the significant difference compared to the control (p < 0.01). (b) Effect of A22 on the delivery efficiency. The pDNA delivery was carried out in the presence of 0, 1, 10 and 100 μg/mL of A22. The concentration of dTat-Sar-EED5 was fixed to 100 μM. The data were normalized by ‘0 μg/mL A22’. Data are the mean ± SD of four cell cultures. Asterisk shows the significant difference compared to the control (without A22) (p < 0.01). Statistically significant differences between samples were determined by the Student t-test.
- Along with the comment, we revised the figure caption of Figure 7b.
“Figure 7. The pDNA delivery into E. coli cells …(b) Effects of dTat-Sar-EED5 concentrations on the transformation efficiency of E. coli. The pDNA delivery was carried out under different concentrations of dTat-Sar-EED5 (0, 1, 10, 50 and 100 μM). The N/P ratio was set to 5.0, where a fixed amount of plasmid DNA (1 μg) was mixed with various amounts of peptide BP100-(KH)9. ‘pDNA only’ means plasmid DNA alone without BP100-(KH)9 and dTat-Sar-EED5. ‘0 µM of dTat-Sar-EED5’ means pDNA complex (BP100-(KH)9/plasmid DNA) without dTat-Sar-EED5. Data were normalized by ‘0 μM dTat-Sar-EED5’. Data are the mean ± SD of three cell cultures.
Comment 3: How is statistical significance calculated in figure 5?
Response: In Figure 5, statistically significant differences between samples were determined by the Student t-test, where a p < 0.01 indicated a significant difference.
- Along with the comment, we added a sentence in the figure caption for Figure 5.
“Figure 5. The pDNA delivery efficiency under various conditions. …difference compared to the control (without A22) (p < 0.01). Statistically significant differences between samples were determined by the Student t-test.”
Comment 4: It would be also quite essential to know the general level of colonies acquired in the most efficient gene transfers (per µg of plasmid or something) i.e. the delivery efficiency. I don’t see that mentioned anywhere.
Response: Yes, we agreed with the reviewer’s comment. The plasmid delivery efficiency varied in each experiment. It maybe because of state of cell cultures. Maximum colony number was 600 per µg of plasmid under the most efficient gene transfer conditions.
- Along with the comment, we added a sentence. (page 5, line 169)
“Overall, the highest plasmid delivery efficiency was 600 cfu per µg of plasmid, which was obtained under condition with N/P ratio 2.0, 100 μM dTat-Sar-EED5, log phase cell culture and overnight recovery period.”
Comment 5: Also, the amount of DNA used in all in the experiments could be mentioned already in the Results and discussion section or at least that it was always fixed amount. This was something I did not understand immediately from the description of the experiments.
Response: Yes, the amount of pDNA was always fixed at 1 μg when preparing the pDNA complex. This was mentioned in the ‘Materials and Methods’ section, ‘3.2 Preparation of peptide-plasmid DNA complex’.
We have added this ‘a fixed amount of pDNA (1 μg)’ description in the Results and Discussion section and also in the figure captions.
- Along with the comment, we revised the text. (page 3, line 104)
“The pDNA complexes were prepared at various N/P ratios ranging from 0.1 to 10 (the ratio of the moles of cationic amine groups of the peptides to those of phosphate groups of the DNA), with a fixed amount of pDNA (1 μg) and analyzed by an electrophoretic mobility assay (Fig. 2a).”
- Along with the comment, we revised the figure captions.
“Figure 2. Characterization of pDNA complexes. …mixed with BP100-(KH)9 at different N/P ratio, with a fixed amount of pDNA (1 μg). Data are the mean ± SD of at least three experiments.”
“Figure 4. Optimization for the pDNA delivery into R. sulfidophilum cells. (a) Effect of N/P ratio on the efficiency. The pDNA complexes was prepared using a fixed amount of pDNA (1 μg) at N/P ratio 0.1, 0.5, 1, 2, 5 and 10. The data…”
Figure 7. The pDNA delivery into E. coli cells. …The N/P ratio was set to 5.0, where a fixed amount of plasmid DNA (1 μg) was mixed with various amounts of peptide BP100-(KH)9 …”
Comment 6: In figure 4 B using the word Preculture is confusing since in the text it is recovery period, if I understood correctly. This could be corrected.
Response: Yes, we agreed with the reviewer’s comment. We apologize for our mistake that had created confusion here. It should be ‘recovery period’.
- Along with the comment, we revised the Figure 4b and figure caption.
Figure 4. Optimization for the pDNA delivery into R. sulfidophilum cells. (a) Effect of N/P ratio on the efficiency. The pDNA complexes was prepared using a fixed amount of pDNA (1 μg) at N/P ratio 0.1, 0.5, 1, 2, 5 and 10. The data were normalized by ‘N/P ratio 0.5’. (b) Effects of recovery period and cell growth stage on the delivery efficiency. The log phase (OD660 of around 1.0) cells and the stationary phase (OD660 of around 4.0) cells were used for experiments. The concentration of dTat-Sar-EED5 was fixed to 100 μM. Data are the mean ± SD of at least three cultures. The data were normalized by ‘o/n recovery period and log phase cells’ condition.
Comment 7: row 253: Only plasmid DNA could not.. à Plasmid DNA alone could not..
Response:
- Along with the comment, we revised the text. (page 7, line 259)
“Plasmid DNA alone could not be delivered into E. coli cells …”
Comment 8: row 54: CPPs have the the ability that translocate.. à CPPs have the the ability to translocate..
Response:
- Along with the comment, we revised the text. (page 2, line 54)
“CPPs have the ability to translocate across the cell membrane and …”
Comment 9: row 211: I would remove the end of the sentence: and the delivery is mediated by actin homolog MreB. This is because definite proof of this is not shown. Also, in other statements thorough out the text this is taken into account.
Response: We have removed the end of sentence ‘… and the delivery is mediated by actin homolog MreB’.
- Along with the comment, we revised the text. (page 6, line 218)
“The dTat-Sar-EED5 induced cellular internalization of pDNA complex into R. sulfidophilum cells was successfully demonstrated in this study. Tat is shown …”
Comment 10: All in all the study is very interesting in terms of genetic manipulation of microbes and the method itself could be one avenue to explore further in bacteria where gene delivery is challenging. Thus, I recommend publication after minor modifications.
Response: Thank you for the encouraging comments and compliments.

Round 2
Reviewer 1 Report
The manuscript is suitable for publication in the current format.